# Time Series Prediction of Sea Surface Temperature Based on an Adaptive Graph Learning Neural Model

**Tingting Wang [1], Zhuolin Li [1], Xiulin Geng [1], Baogang Jin [2,\*] and Lingyu Xu [1,3,\*]**

[1]  Department of Computer Engineering and Science, Shanghai University, Shanghai 200444, China; wtt_forsth@163.com (T.W.); lzl_shu@163.com (Z.L.); gxl_shu@163.com (X.G.)
[2]  Beijing Institute of Applied Meteorology, Beijing 100029, China
[3]  Shanghai Institute for Advanced Communication and Data Science, Shanghai University, Shanghai 200444, China
\*  Correspondence: jinbaogang2021@163.com (B.J.); xly@shu.edu.cn (L.X.)

**Abstract:** The accurate prediction of sea surface temperature (SST) is the basis for our understanding of local and global climate characteristics. At present, the existing sea temperature prediction methods fail to take full advantage of the potential spatial dependence between variables. Among them, graph neural networks (GNNs) modeled on the relationships between variables can better deal with space–time dependency issues. However, most of the current graph neural networks are applied to data that already have a good graph structure, while in SST data, the dependency relationship between spatial points needs to be excavated rather than existing as prior knowledge. In order to predict SST more accurately and break through the bottleneck of existing SST prediction methods, we urgently need to develop an adaptive SST prediction method that is independent of predefined graph structures and can take full advantage of the real temporal and spatial correlations hidden indata sets. Therefore, this paper presents a graph neural network model designed specifically for space–time sequence prediction that can automatically learn the relationships between variables and model them. The model automatically extracts the dependencies between sea temperature multivariates by embedding the nodes of the adaptive graph learning module, so that the fine-grained spatial correlations hidden in the sequence data can be accurately captured. Figure learning modules, graph convolution modules, and time convolution modules are integrated into a unified end-to-end framework for learning. Experiments were carried out on the Bohai Sea surface temperature data set and the South China Sea surface temperature data set, and the results show that the model presented in this paper is significantly better than other sea temperature model predictions in two remote-sensing sea temperature data sets and the surface temperature of the South China Sea is easier to predict than the surface temperature of the Bohai Sea.

**Keywords:** time series; deep learning; graph structure learning; spatial-temporal graph; prediction; sea surface temperature (SST)

## 1. Introduction

The study and understanding of space–time distribution and changes in seawater temperature is an important aspect of oceanography, and it is of great significance to marine fisheries, aquaculture, and marine operations [1,2]. Sea surface temperature (SST) is a key parameter for measuring ocean thermal energy, and it also has a significant impact on regional climate change. For example, the seasonal prediction of high temperature anomalies in the eastern United States was improved by studying the evolution mode of SST anomalies; seasonal surface temperature anomalies in Europe were improved by studying SST anomalies in northern European waters; and the El Niño-Southern oscillation (ENSO) over the equatorial eastern Pacific can be effectively predicted by studying the variation law of SST [3–6]. Therefore, the accurate forecasting of SST is the basis for our

understanding of local and global climate characteristics. However, the ocean not only has independent tidal and ocean current systems, but also multi-dimensional information, complex spatio-temporal correlations, a large area, a multi-model, remote correlation, and other problems, which cause great difficulties for the prediction and mechanisms of discovery of SST.

At present, the prediction methods of SST time series data can be divided into three categories. The first is a numerically-based approach, which predicts ocean elements according to a set of predefined rules such as those of coupled ocean-atmosphere models (GCMS) [7–9]. However, these methods not only require extremely high computational resources and a professional knowledge of thermodynamics [10], but also involve complex external data, which requires a large amount of model start-up time and set of integral assumptions. In addition, because these methods need to predict many parameters at the same time, it is a difficult task to accurately predict a single parameter for this method [11]. Secondly, some methods of machine learning technology show a great performance in the prediction of complex time series, for example, KNN [12] for ocean current prediction and SVR [13] for wind field prediction. However, in the process of prediction, these methods only consider the temporal correlation of the data and ignore the spatial information, which leads to the failure of the model to effectively capture the spatio-temporal-dependent information of nonlinear correlation. Thirdly, with the development of deep learning, more and more researchers have begun to use neural networks to predict ocean elements. A lot of methods based on neural networks are widely applied in predicting sea surface temperature [14]. Zhang Q et al. [5] proposed a fully connected network model (FC_LSTM) based on LSTM for sea surface temperature prediction. Xie J et al. [15] built a GED model, composed of GRU and an attention mechanism, which can be used to predict multi-scale SST. However, the above methods regard SST prediction as a single-source time-series prediction problem and do not make full use of the hidden relationship between time and space. This will not only lead to a failure to learn the fusion mode of multi-elements in the actual complex marine environment, but also cause the loss of fusion information and a decline in prediction accuracy.

At the same time, the graph neural networks (GNNs) have made great progress in dealing with relation dependence. GNNs can make full use of the relationship between variables, especially in the spatio-temporal correlation, due to the characteristics of graph neural networks of replacement invariance, local connectivity, and composition. Existing graph neural network methods usually use GCN-based methods to model unstructured sequences and the inter-dependencies of different variables such as ASTGCN [16], STS-GCN [17], and GMAN [18]. These methods take multivariate time series and externally predefined graph structures as inputs to predict future values or labels of multivariate time series. Compared with the previous methods, these methods have made significant improvements. However, these methods still cannot be used for SST time series because of the following difficulties:

- Graph structure learning method: At present, most GNNs implement spatio-temporal series prediction based on predefined graph structure, but there is no graph structure displayed in SST time series data. The relationship between spatial points in the SST data set is hidden in the data, which needs to be mined instead of existing as prior knowledge. Therefore, how to mine the relationship between variables from the SST data and learn graph structure by deep learning remains a big challenge at present.
- End-to-end framework: At present, most GNNs only update the hidden state of input sequence data in the learning process, neglecting to update the graph structure in time. Therefore, how to learn the graph structure and time series data simultaneously in a unified end-to-end framework is also a challenge.

To solve the above problems, this paper proposes a graph neural network model (AGLNM) specially designed for spatio-temporal series prediction, which can automatically learn and model the relationship between variables. The AGLNM mainly consists of a graph learning module, graph convolution module, and time convolution module.

The contribution points of this paper are as follows:

- The graph learning module designed by this paper breaks the current limitation of GNN application in SST data sets without an explicit graph structure; the module can not only mine the hidden spatial-temporal dependencies in SST sequential data, but also process SST data without a predefined graph structure by automatically learning the graph structure.
- This paper proposes an end-to-end framework which includes a graph learning module, graph convolution module, and time convolution module. In this framework, a graph loss mechanism is added to guide the graph structure to update to the optimal direction according to downstream tasks, which makes the final graph structure effectively aid in SST prediction. Compared with other models, the mean absolute error of this model is reduced by more than 13%, and it can be transplanted to data sets without graph structure.
- In this paper, AGLNM is evaluated for the first time on two remote-sensing SEA surface temperature data sets, and compared with several representative time series prediction models. The experimental results show that the performance of AGLMN is better than other advanced models.

The rest of this article is organized as follows. In Section 2, we formulate the question and presents the details of the proposed AGLNM. In Section 3, we evaluate the performance of the AGLNM and analyze the experimental results. Finally, Section 4 gives our conclusions.

## 2. Materials and Methods

### 2.1. Problem Description

This paper mainly studies the prediction of sea surface temperature time series data. Since the SST data itself has spatial information, we can divide the SST data set into grid data according to latitude and longitude. Then, we regard the SST data set as a grid data set S composed of multi-variable time series X, where X represents the time series data set under different latitude and longitude, and N represents the number of grids after latitude and longitude division of the data set. For each time series data X, a training window with time step P and a prediction window with time step Q are given. Our purpose is to find a mapping function which could predict the SST sequence Y at the future time according to the SST sequence X at the past time. The X, Y, and the mapping function F are defined as follows:

$$F(X, \theta) = Y \tag{1}$$

$$X = \left\{ X_{t_1}, X_{t_2}, X_{t_3}, \cdots, X_{t_P} \right\} \tag{2}$$

$$Y = \left\{ X_{t_{P+1}}, X_{t_{P+2}}, X_{t_{P+3}}, \cdots, X_{t_{P+Q}} \right\} \tag{3}$$

where $X_{t_1} \in R_N$ represents the temperature value of the time series data at $t_i$ moment, and P and Q represent the length of the historical series and the predicted series, respectively.

In this paper, $G = (V, E)$ is used to represent the graph formed on sea surface temperature data, where V is the set of spatial nodes of SST information, E is the set of associated relations between spatial nodes, and N is used to represent the number of spatial nodes in the graph. In addition, A represents the collar matrix of the relation between the spatial nodes V in the graph, specifically expressed as $A \in R_{N \times N}$ with $A_{ij} = c > 0$ if $(v_i, v_j) \in E$ and $A_{ij} = 0$ elsewise.

### 2.2. Method Overview

Figure 1 briefly describes the end-to-end framework structure of the proposed approach, which is called the adaptive graph learning network model (AGLNM). The model framework mainly includes a graph learning module, a graph loss mechanism, graph convolution modules, and time convolution modules. The graph learning module can mine the adaptive adjacency matrix from the data, discover the hidden association between

nodes, and then serve as the input for the graph convolution module. The graph loss mechanism can continuously update and optimize the adaptive adjacency matrix to the real dependence between the spatial points hidden in the SST data. The graph convolution module can be used to capture the dependencies between SST spatial points. The time convolution module is used to mine the time series pattern corresponding to each space point. In addition, residual links are added before and after each pair of spatio-temporal convolution modules to avoid the problem of gradient disappearance. The output of the final model will project the hidden correlation features of the SST time series into the output dimensions of the desired prediction series. Each module of the model is explained in detail in the following sections.

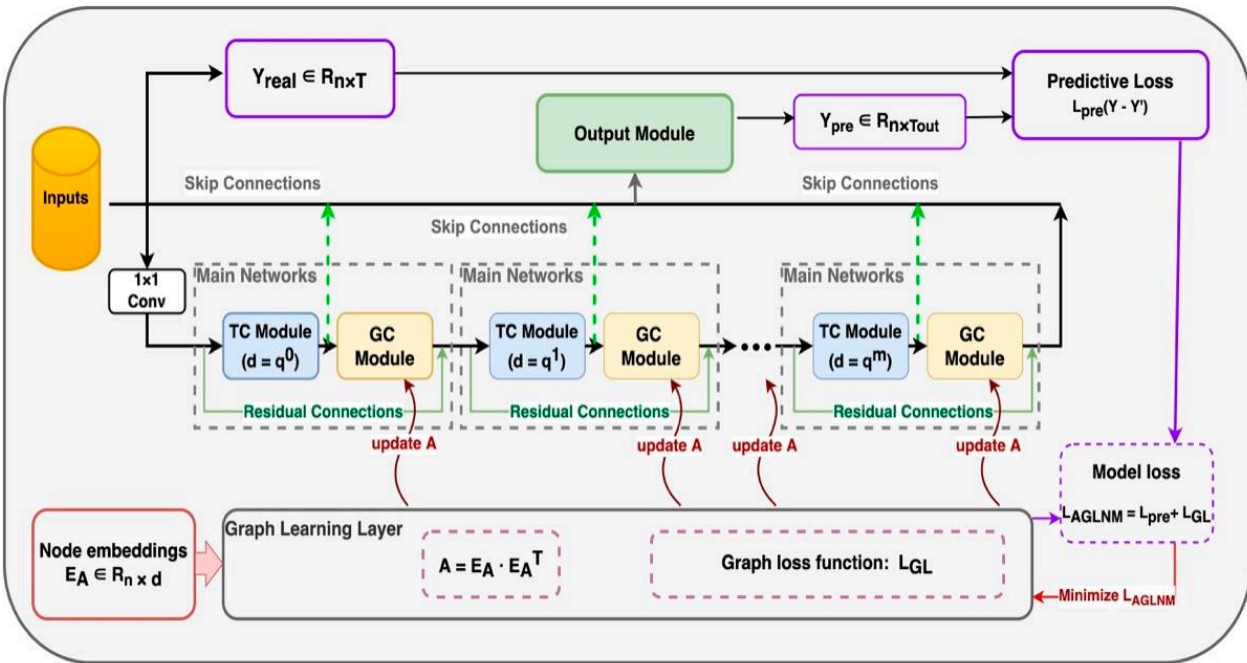

**Figure 1.** Diagram of the overall framework of the adaptive graph learning neural model (AGLNM). TC module represents the time convolution module, while GC module is the graph convolution module. $Y_{real}$ represents the real SST value sequence, $Y_{pre}$ represents the predicted SST value sequence, $L_{pre}$ represents the loss function of SST prediction, $L_{GL}$ represents the graph loss function of the graph learning layer, and $L_{AGLNM}$ represents the composite loss function of the model AGLNM. $E_A$ is the SST space node embedding, and A represents the adaptive adjacency matrix.

### 2.3. Adaptive Graph Learning Network (AGLN)

The graph learning module is designed to mine adaptive adjacency matrices driven by the data. However, in terms of time series prediction, most of the existing graph neural network methods based on mining adjacency matrices rely heavily on predetermined graph structure and cannot update graph structure over time during training, that is, adjacency matrix A needs to be calculated according to a distance function or similarity function before input to the model. Firstly, this method of calculating violence requires a great deal of domain knowledge. Secondly, predefined graph structures containing only explicit spatial information cannot mine hidden spatial dependencies for this prediction task, which may lead to considerable bias in the final prediction. Finally, predefined graph structures are not portable or compatible with other prediction tasks. To solve the above problems, the graph learning module proposed herein is an adaptive graph learning network (AGLN) driven by raw data, which is specifically used to automatically mine the hidden interdependence in

real data. The adjacency matrix A containing the graph structure information is calculated by Formula (4). Specific instructions are as follows:

$$\widetilde{A} = D^{-\frac{1}{2}} A D^{-\frac{1}{2}} = Softmax\left(ReLU\left(E_A \cdot E_A^T\right)\right) \tag{4}$$

where $A$ is the adjacency matrix, $D$ is the degree matrix, $E_A \in R_{N \times d}$ represents the node embedding matrix randomly initialized by AGLN for all nodes which can be learned and updated through the training process, $N$ represents the number of nodes or spatial points, and $d$ represents the dimension of node embedding. The transition matrix $\widetilde{A}$ is the adaptive matrix obtained after the normalization of adjacency matrix $A$ by the softmax function. It is worth noting that we directly update and calculate the dot product $\widetilde{A}$ of adjacency matrix $A$ and Laplace matrix $L$, instead of generating adjacency matrix $A$ and Laplace matrix $L$ separately, which can reduce the computational overhead in the iterative training process.

In addition, the AGLN adds the following graph loss mechanism to continuously update and optimize the adjacency matrix $A$ toward the real spatial dependence of the SST data.

$$L_{GL} = \sum_{i=0,j=0}^{N} \|x_i - x_j\|_2^2 \widetilde{A}_{ij} + \gamma \|\widetilde{A}\|_F^2 \tag{5}$$

where $\widetilde{A}_{ij} \in R_{N \times N}$ represents the spatial dependence relation between node $i$ and node $j$, $\|x_i - x_j\|_2^2$ is the calculation formula for the dependence relation between space point $i$ and $j$. The smaller the value is, the larger the value of transition matrix $\widetilde{A}_{ij}$ will be. Due to the simple property of transition matrix $\widetilde{A}$, the second term in the formula can control the sparsity of the learned adjacency matrix $A$.

It can be seen from Formula (5) that $L_{GL}$ is a graph loss function driven by the spatial node data. Therefore, minimizing $L_{GL}$ enables the AGLNM to adaptively mine the real spatial correlation hidden in SST data. However, minimizing the value of the graph loss function $L_{GL}$ alone may only provide a general solution. Therefore, we used $L_{GL}$ as the regularization term in the final loss function of this paper to participate in the training. The node-embedding matrix $E_A$ captures the hidden spatial dependencies between different nodes through automatic updating in training, and finally generates the adaptive adjacency matrix, which is then used as the input of the next graph convolution network.

### 2.4. Time Convolution Module

The time convolution module contained in the AGLN model proposed in this paper mainly adopts a gated structure and dilated convolution. The gated structure extracts multiple time patterns from the data by adopting multiple convolution modes in each convolution layer so as to effectively control the information flow. By controlling the expansion coefficient, dilatation convolution can enable the model to process longer time series in SST prediction tasks so as to better mine the hidden time correlation of SST data.

Figure 2 is the structural schematic diagram of the time convolution module. We design four one-dimensional convolution filters of different sizes as the initial layer of the time convolution module, which can extract the sequence patterns contained in the SST time series data. Then, the tangent hyperbolic activation function, sigmoid activation function, and a gating device are used to control the amount of information transmitted to the downstream task.

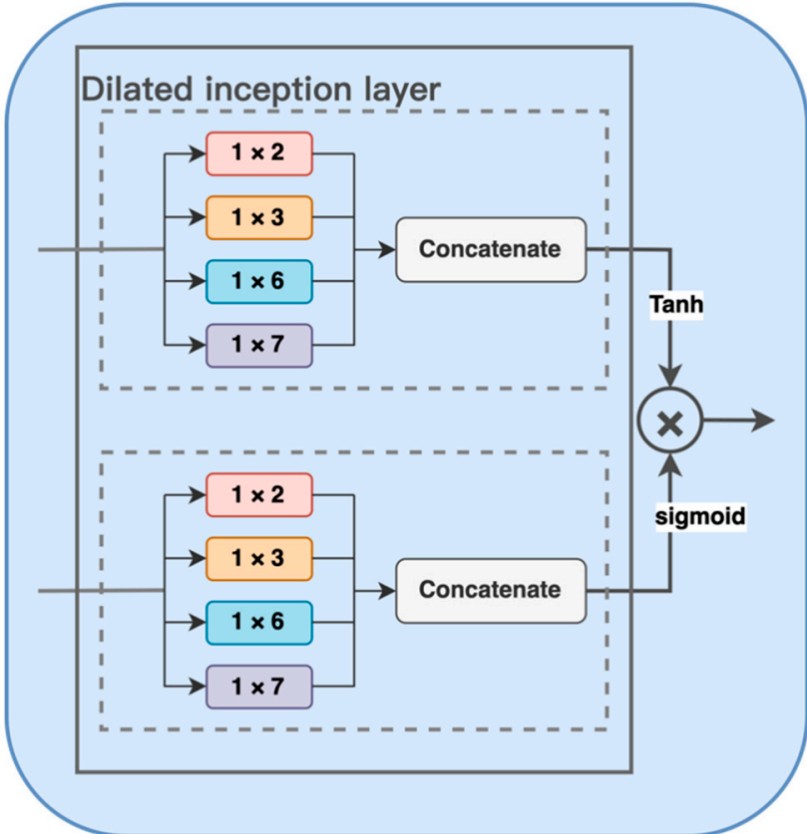

**Figure 2.** Structure diagram of temporal convolution layer. Tanh and sigmoid denote the tangent hyperbolic activation function and sigmoid activation function, respectively.

First of all, in order to simultaneously capture the long and short term signal patterns in SST data, we consider using multiple filters of different sizes to form an initial dilated convolution layer. It is worth noting that we need to choose the appropriate filter size to cover several inherent periods of the time series, such as 3, 7, 9, and 30. According to the periodicity of SST data changes, four filters with sizes of $1 \times 2$, $1 \times 3$, $1 \times 6$ and $1 \times 7$ are selected as a set of standard extended one-dimensional expansion convolution layers.

Secondly, in order to enable the model to better deal with long time series data in SST prediction tasks, we need to select an appropriate expansion coefficient. In standard convolutional networks, the field of view increases with the network depth and the number of convolutional kernels. However, SST time series prediction needs to deal with a large field of vision. If we use standard convolution, we have to design a very deep network or a very large filter, which leads to an explosion in model complexity and computation cost. Therefore, in order to avoid such a situation, we use the expansion coefficient $dn$ to increase the field of view and reduce the computational complexity of the model by changing the input down-sampling frequency. For example, when the expansion coefficient $dn$ is one, the field size $V$ of the dilated convolutional network can be calculated by Formula (6).

$$V = \frac{(k-1)\left(q^d - 1\right)}{(q-1)} + 1 \tag{6}$$

where $V$ is the size of the field of vision of the convolutional network, $d$ is the depth of the convolutional network, $k$ is the size of the convolution kernel or the size of the filter, and $q$ is the growth rate of the expansion coefficient, $q > 1$.

As can be seen from Formula (6), in the case of the same number of deep convolution kernels in the network, compared with the standard convolution field with linear growth, the field of field of empty convolution can grow exponentially with the depth of the

network. In this way, a longer time series pattern can be captured by processing a larger field. According to the above principle, the initial output of the expansion convolution layer can be calculated by Formula (8).

$$X \times f_{1 \times k}(t) = \sum_{s=0}^{k-1} (s) X(t - dn \times s) \qquad (7)$$

$$Z = \text{concat}(X \times f_{1 \times 2}, X \times f_{1 \times 3}, X \times f_{1 \times 6}, X \times f_{1 \times 7}) \qquad (8)$$

where $X \in R^T$ is the one-dimensional sequence input, $s$ is the step size, $t$ is the time, $dn$ is the expansion coefficient, and $f_{1 \times 2}$, $f_{1 \times 3}$, $f_{1 \times 6}$, and $f_{1 \times 7}$ are four filters of different sizes. The module truncates the output of the four filters to the same length as the largest filter and then connects a set of filters across the channel dimension to output $Z$.

*2.5. Graph Convolution Module*

As described in Section 2.1 regarding the principle of adaptive graph generation, the function of the graph convolution module is to capture the spatial features of nodes with known node structure information. Kipf et al. proposed a first-order approximation algorithm that smoothens node spatial information by aggregation and transformation of adjacent information of nodes and the algorithm-defined graph convolution layer as in Formula (9) [19,20].

$$Z = \widetilde{A} X W \qquad (9)$$

where $\widetilde{A} \in R_{N \times N}$ represents the adaptive adjacency matrix or transition matrix, $X \in R_{N \times D}$ represents the input time series, $Z \in R_{N \times M}$ represents the output prediction time series, $W \in R_{D \times M}$ represents the model parameter matrix, $D$ represents the data input dimension, and $M$ represents the number of layers of the graph convolution module.

The graph convolution layer can also extract node space features based on the local structure information of the graph. Li et al. proposed a spatio-temporal model containing a diffusion convolution layer by modeling the diffusion process of graph signals and proved the effectiveness of a diffusion convolution layer in predicting road traffic flow sequences [21]. According to the form of Formula (9), the diffusion convolution layer is defined as in Formula (10) [21].

$$Z = \sum_{c=0}^{c} P^c X W_k \qquad (10)$$

where $c \in R$ represents a finite step, and $P^c$ represents the power series of the transition matrix $\widetilde{A}$. Because SST time series data belong to undirected graphs, we propose the graph convolution layer formula as shown in Formula (11) by combining the time dependence and hidden dependence of adaptive graph learning layer in this paper.

$$Z = \sum_{c=0}^{c} \widetilde{A}^c X W_c \qquad (11)$$

where $\widetilde{A}$ is the adaptive adjacency matrix, $X \in R_{N \times D}$ represents the input history time series, $Z \in R_{N \times M}$ represents the output prediction time series, and $W \in R_{D \times M}$ represents the model parameter matrix. In this paper, graph convolution belongs to a space-based method. Although the graph signal and node feature matrix are used interchangeably in this paper for consistency, the graph convolution represented by the above formula is still interpretable and can aggregate transform feature information from different neighborhoods.

## 3. Experiments

*3.1. Data Set*

We adopted NOAA's earth system research laboratory provided by the physical sciences, NOAA and high resolution SST data of the high resolution ocean surface temperature data set (OISST) as the source of experiment data (https://www.esrl.noaa.gov/psd/ ac-

cessed on 4 May 2022). The data set covers the mean daily, weekly, and monthly sea surface temperatures of the global ocean from September 1981 to the present, which is updated over time. The daily mean SST data set (4018 days) with a spatial resolution of $0.25° \times 0.25°$ from January 2008 to December 2013 in the Bohai Sea and the South China Sea was selected as the data set for the experiment in this paper. We first preprocessed SST data from the Bohai Sea (135 points) and South China Sea data sets (2307 points), and then divided the data sets into training data sets and test data sets as shown in Table 1 for daily mean SST prediction. Firstly, the Bohai sea data set is trained and tested. Then, in order to verify the robustness of the model, this paper also carries out comparative experiments of six models on the South China Sea data set.

**Table 1.** The statistics of data sets.

| Sea Area | Spatial Coverages | Purpose | Temporal Coverages |
|---|---|---|---|
| Bohai Sea (1/4° latitude/1/4° longitude) | 37.07 N to 41.0 N, 117.35 E to 121.10 E | train test | 1 January 2008 to 26 November 2012 27 November 2012 to 31 December 2013 |
| South China Sea (1/4° latitude/1/4° longitude) | 6.5 N to 21.5 N, 112.5 E to 119.5 E | train test | 1 January 2008 to 26 November 2012 27 January 2012 to 31 December 2013 |

*3.2. Baselines*

In this section, we compare our model with the baselines on the two data sets. The four baselines are as follow:

CGMP [22]: Based on a convolution gated cyclic unit (GRU) and a multi-layer perceptron, the model can not only capture neighbor influence effectively in the spatial dimension, but also process historical information effectively in the temporal dimension.

FC-LSTM [5]: The model is composed of an LSTM layer and a full connection layer. It can apply recursive neural network to SST prediction.

SVR [13]: Support vector regression is widely used in time series prediction.

CFCC-LSTM [11]: The model is composed of a fully connected LSTM layer and a convolution layer, which can combine temporal and spatial information well to predict SST time series data.

GED [15]: GED is a model with a GRU encoder-decoder with SST code and a dynamic influence link (DIL).

*3.3. Experiments Settings and Metrics*

The robustness of AGLNM, CGMP, FC-LSTM, SVR, CFCC-LSTM, and GED were compared by using the open-source deep learning tool PyTorch and the open-source machine learning tool LibSVM on SST data sets of the Bohai Sea and South China Sea, respectively. The six models were trained and tested on the daily mean SST data set to predict the future SST of 1, 3, and 7 days for each spatial point. Specifically, Adam was selected as the optimization algorithm of the model and the learning rate was initialized to 0.01 with a decay. The batch size of the input data was set to 128 and the number of iterations was set to 1000. The mean square error (MSE) and mean absolute error (MAE) were selected as the evaluation indexes of prediction performance. The smaller the MSE or MAE, the better its performance. MSE, MAE, and experimental losses are defined as follows:

$$\text{MSE}(Y, Y') = \frac{1}{l} \sum_{i=0}^{l} (y_i - y_i')^2 \tag{12}$$

$$\text{MAE}(Y, Y') = \frac{1}{l} \sum_{i=0}^{l} |y_i - y_i'| \tag{13}$$

$$L_{AGLNM} = L_{Pre}(Y - Y') + L_{GL} \tag{14}$$

where $Y$ is the predicted future SST series, $Y'$ is the corresponding observed SST series, $y_i$ is the SST series contained in the prediction window, $y_i'$ is the observed SST series corresponding to the window, and l is the total predicted length. In Formula (14), the model adopts the sum of the predicted loss $L_{Pre}(Y - Y')$ and graph loss $L_{GL}$ as the combined loss to participate in the training. The data-driven training model loss can simultaneously learn the association relation of the SST sequence and the graph structure most suitable for the current data set.

*3.4. Results*

3.4.1. Performance of AGLNM on Bohai Sea Data Set

The performance of the proposed AGLN model was compared with CGMP, FC-LSTM, CFCC-LSTM, GED, and SVR by multi-scale prediction on the daily mean SST data set (4018 days) of the Bohai Sea (135 points); that is, predicting the mean SST for the next 1, 3, and 7 days. The experimental results are shown in Table 2. The smaller the MAE and MSE, the better the prediction performance of the model. In order to show the advantages of the AGLNM more clearly, we bold the minimum MSE and minimum MAE in the table. It can be seen that the AGLNM proposed in this paper is superior to the other models at different prediction scales to varying degrees, followed by CGMP. In addition, the MAE of the AGLNM was 0.04, 0.06, 0.1 (13%, 12%, 15%) ahead of the CGMP model with the second highest performance when the prediction scale was 1 day, 3 days, and 7 days, respectively. When the prediction scale was larger, the performance of the AGLNM proposed in this paper had more obvious advantages compared with the other models, which indicates that the data-driven AGLNM can indeed mine and learn the hidden association relationship between spatial nodes in the SST data of the Bohai Sea to a certain extent.

**Table 2.** Prediction results of the Bohai Sea data set (135 points).

| Models | Metrics | Prediction Length (Day(s)) | | |
|---|---|---|---|---|
| | | **1** | **3** | **7** |
| CGMP | *MSE* | 0.1714 | 0.4173 | 0.7449 |
| | *MAE* | 0.3014 | 0.4685 | 0.6301 |
| FC-LSTM | *MSE* | 0.187 | 0.4708 | 0.7985 |
| | *MAE* | 0.3091 | 0.4937 | 0.6574 |
| CFCC-LSTM | *MSE* | 0.1715 | 0.4219 | 0.8233 |
| | *MAE* | 0.3015 | 0.4794 | 0.6771 |
| GED | *MSE* | 0.1723 | 0.4933 | 0.8502 |
| | *MAE* | 0.2956 | 0.5034 | 0.6810 |
| SVR | *MSE* | 0.4716 | 0.6919 | 1.0047 |
| | *MAE* | 0.5058 | 0.6140 | 0.7416 |
| AGLNM | *MSE* | **0.1378** | **0.3057** | **0.5318** |
| | *MAE* | **0.2629** | **0.4031** | **0.5398** |

3.4.2. Performance of AGLNM on the South China Sea Data Set

In order to verify the robustness of the AGLNM, this paper compared the above six models on the SEA surface temperature daily data set (4018 days) in the South China Sea (2307 points), and the experimental results are shown in Table 3. First of all, we can see that the performance of the AGLNM in this paper still maintains a great advantage compared with the other methods when the prediction scale is 1 and 3 days. When the prediction scale was 7 days, the AGLNM was slightly behind the CGMP model with a gap of less than 0.01, but it was still slightly ahead of the FC-LSTM and still ahead of the CFCC-LSTM, GED, and SVR models by a large margin. Therefore, the AGLNM still maintained the best overall performance, followed by the CGMP. Secondly, the mean absolute error of temperature prediction was in the range of 0.2, which is the international leading level.

However, the MAE can still outperform the CGMP model ranked second by 0.026 (14.6% difference) at the prediction scale of 1 day. The occurrence of the fault phenomenon reflects that the performance of the AGLNM designed in this paper is significantly superior to other models in nature.

**Table 3.** Prediction results of the South China Sea data set (2307 points).

| Models | Metrics | Prediction Length (Day(s)) | | |
|---|---|---|---|---|
| | | **1** | **3** | **7** |
| CGMP | *MSE* | 0.0613 | 0.1058 | **0.1658** |
| | *MAE* | 0.1782 | 0.2411 | **0.3099** |
| FC-LSTM | *MSE* | 0.0765 | 0.1387 | 0.1721 |
| | *MAE* | 0.1792 | 0.2788 | 0.317 |
| CFCC-LSTM | *MSE* | 0.1057 | 0.1234 | 0.1839 |
| | *MAE* | 0.2502 | 0.2663 | 0.3531 |
| GED | *MSE* | 0.0818 | 0.1402 | 0.2460 |
| | *MAE* | 0.2146 | 0.2823 | 0.3881 |
| SVR | *MSE* | 0.0953 | 0.1569 | 0.2416 |
| | *MAE* | 0.2233 | 0.2955 | 0.3726 |
| AGLNM | *MSE* | **0.0485** | **0.1057** | 0.1907 |
| | *MAE* | **0.1520** | **0.2292** | 0.3156 |

## 4. Discussion

In order to display and discuss the experimental results in this paper in a more convenient and vivid way, we plotted the experimental results MAE and MSE with the prediction scale of 1 day, 3 days, and 7 days, respectively, on the Bohai sea data set and the South China Sea data set of the six models in Figure 3 for further comparison.

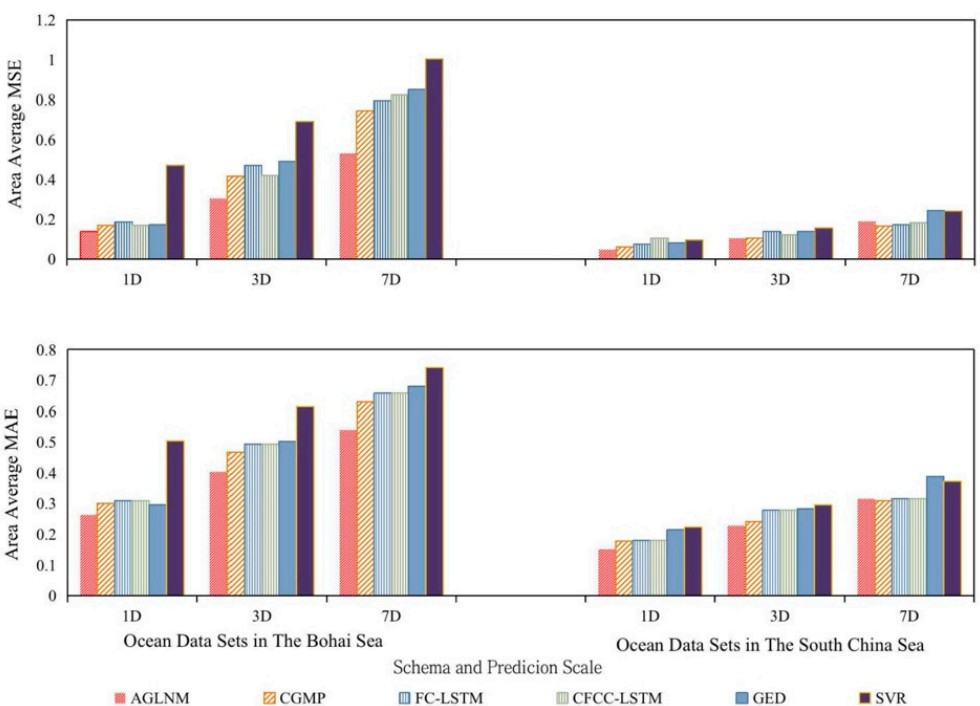

**Figure 3.** SST prediction results of the six models at different scales and sea areas.

### 4.1. Differences between Bohai Sea and South China Sea Data sets

By comparing the performance of each model in different sea areas seen in Figure 3, we can find some similarities of laws. Under the same prediction scale, the prediction index values of each model in the Bohai Sea data set are significantly higher than those in the South China Sea data set. The experimental results show that the SST of the South China Sea is easier to predict than that of the Bohai Sea.

Located in tropical and subtropical regions, the South China Sea is characterized by significant tropical maritime climate and small seasonal temperature variations, resulting in the weaker temporal and spatial correlation of the SST data in the South China Sea. Because the Bohai Sea is located in the north temperate zone, it is greatly affected by temperate monsoons, continental climate, ocean currents and other factors, so the SST data will show a stronger spatial correlation. Through practical analysis, it is more difficult to predict the SST in the Bohai Sea area affected by more factors, which is consistent with the experimental results in this paper.

### 4.2. Advantages of AGLNM

By comparing the performance of the AGLNM with other models, it can be seen from Figure 3 that the experimental performance of the adaptive graph learning network model proposed in this paper is obviously better than other models, especially in the Bohai sea data set with a stronger spatial correlation, regardless of the strength of the spatial correlation in the data set. This shows that the AGLNM proposed in this paper can not only adapt to different data sets, but can also mine well and utilize the spatial dependence of SST.

### 4.3. Limitations of AGLNM

The defects of the AGLNM are also obvious. As can be seen from the structure of the model in Figure 1, the AGLNM needs to constantly update the graph structure in order to make the graph structure learned by the model more consistent with the real association relationships contained in the data set. The update of the graph structure requires a recalculation of the adjacency matrix and the new adjacency matrix needs to be added to the next update, which obviously consumes more computing resources and more training time. Therefore, this model is not suitable for small data sets.

As can be seen from Figure 3, AGLNM has a small space to improve the prediction performance of the South China Sea data set, but it requires more computing time and resources. This may be because the South China Sea data set is not subject to seasonal fluctuations and the feature changes are small and stable. For this kind of data set, AGLNM has little effect on graph learning, which leads to a low model efficiency.

## 5. Conclusions

Most of the existing SST prediction methods fail to fully mine and utilize the spatial correlation of SST, and most of the graph neural networks which model the variable relationship rely heavily on the predefined graph structure (i.e., use prior knowledge to construct the spatial point dependence). To solve the above problems, this paper specially designed an end-to-end model AGLNM for SST prediction without explicit graph structure, which can automatically learn the relationship between variables and accurately capture the fine-grained spatial correlations hidden in sequence data.

The experimental results of the performance test on the Bohai sea and South China Sea SST data sets show that: Firstly, AGLNM can effectively capture the dependence relationship between ocean spatial points. Secondly, the overall performance of the AGLNM is significantly better than that of the CGMP, FC-LSTM, CFCC-LSTM, GED, and SVR models in different sea areas and at different prediction scales. Finally, under the same prediction scale, the SST of the South China Sea is easier to predict than that of the Bohai Sea.

The AGLNM proposed in this paper has a better portability and can self-mine the hidden spatial association relationship contained in the data set that is the most consistent

with the characteristics of the real data and can be better applied to large and complex data sets in the future. Based on the advantages and disadvantages of the AGLNM, the model can be better applied to data sets with more complex environments, large feature fluctuations, and stronger time-space correlations, such as the data set of the first island chain affected by monsoons, ocean currents, and man-made operations simultaneously, which can make full use to the advantages of the model and have stronger military significance.

**Author Contributions:** Conceptualization, T.W. and Z.L.; methodology, Z.L.; software, T.W.; validation, Z.L., B.J. and L.X.; formal analysis, L.X.; investigation, B.J. and L.X.; resources, B.J. and L.X.; data curation, T.W.; writing—original draft preparation, T.W.; writing—review and editing, X.G.; visualization, T.W.; supervision, B.J. and L.X.; project administration, B.J. and L.X.; funding acquisition, B.J. and L.X. All authors have read and agreed to the published version of the manuscript.

**Funding:** This research was funded by the National Program on Key Research Project of China, grant number 2016YFC1401900.

**Institutional Review Board Statement:** Not applicable.

**Informed Consent Statement:** Not applicable.

**Data Availability Statement:** Publicly available data sets were analyzed in this study. This data can be found here: https://psl.noaa.gov/ accessed on 3 May 2022.

**Conflicts of Interest:** The authors declare no conflict of interest.

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
