# Peer review of "Time Series Prediction of Sea Surface Temperature Based on an Adaptive Graph Learning Neural Model"

_futureinternet, doi:10.3390/fi14060171_

Round 1

Reviewer 1 Report

In this paper, the authors develop a GNN-based method to predict sea surface temperatures from data. The results show that the proposed method can achieve better accuracy than the selected reference methods.

  1. One aspect that requires clarification is the relationship between the proposed method and the current and future Internet.
  2. In page 4, line 162, a suggestion is to be specific that the "existing methods" refers to those applied to the problem scope. The topic of mining adjacency data, i.e., link prediction, is not an uncommon application of GNNs.
  3. A key benefit of the proposed method refers to its inference capabilities. From the description, it is not fully clear whether the model was trained and tested separately using the 2 datasets, Bohai Sea and South China Sea,  or using both sets at the same time. It would be especially interesting to see how well training on one set translates into predicting values for the second set.
  4. It will be useful to report the dataset sizes and the training time.
  5. Various writing improvements are needed. The space before references to the bibliography has many times been omitted. Various typos need correction, e.g., "Gragh" in 2.5. Also, acronyms have been used without definition

Author Response

I'm very sorry to have kept you waiting. Due to the epidemic, I was suddenly isolated for 7 days and had no computer to modify the manuscript. As a result, the discussion and modification of the manuscript were postponed. And Thank you very much for your understanding.

Reviewer 2 Report

Dear authors,

the paper starts off very well, Introduction and methodology are well structured and written. However, Results and Discussion are weak and should be improved. There are mistakes (3.2 is completely missing), and Discussion starts directly with a figure, not even one sentence mentioning it. Also the conclusions could be strengthened.

I will have no objection accepting it once it looks concise and well written throughout.

The forecasting of sea level temperature with the use of a derivative of graph neural networks. Graph neural networks are a relevant issue that will become even more popular in time and seems that the approach is good, but the use case is neutral to me in terms of interest.

The topic (in terms of algorithms and proposed methodology) is original, at least it is proposing a new variant of graph neural networks which is always good for research. The subject area is well covered and it shows in the introduction, where other papers are presented. So overall, yes, it adds to the subject area of GNN.

First two sections are well written, the rest is sloppy. Mistakes like not including text in the beginning of a section and starting with un-referenced figures.

The conclusions are consistent and address the main question based on the results, however, they could benefit from an expansion and elaboration.

In general to me it feels that the paper has been written by two different people, one that wrote the introduction and methodology and one that wrote the results and discussions. The latter sections are sloppy in contrast to the first two that are written in a well-structured academic manner.

Author Response

I'm very sorry to have kept you waiting. Due to the epidemic, I was suddenly isolated for 7 days and had no computer to modify the manuscript. As a result, the discussion and modification of the manuscript were postponed. And thank you very much for your understanding.

Reviewer 3 Report

Very interesting and current manuscript. In the introduction, the need to solve the problem is well presented, together with a number of studies that have addressed this topic, and an overview of methods that are important for solving this problem. However, more quality and up-to-date resources could be used. The section on the description of the methods used is described in great detail, and I consider this to be the best part of the paper. The results are potentially applicable and important. The article is very well written, but still needs some editing:

  1. Abbreviations could be explained (for example from Figure 1).
  2. The information on the data set used could rather be the subject of a methodology.
  3. In the Data Set, you state that the data "… from September 191 to the present" was used, but it is not clear what "present" is. You must specify a specific time period.
  4. It is probably not good to mention Chapter 3. Results and Chapter 3.5 Results, the subchapter should be named differently.
  5. To support the scientific nature of the article, it would be good to set research questions or rather hypotheses. There is also a lack of a strict aim of the paper.
  6. Figure 4 should be preceded by some text that introduces it. Why is subchapter 4.4.1 and 4.4.2 and not just 4.1 and 4.2?
  7. The conclusion should include information on the limits of the research, suggestions for continuing the research and more detailed information on the practical consequences of the results of your study.
  8. What software did you use for the calculations?

Author Response

(The authors gave the same response as above.)

Round 2

Reviewer 2 Report

The manuscript is clearly worked on and improved. I have no objection towards its publication.